# Predicting cancer origins with a DNA methylation-based deep neural network model

**Chunlei Zheng, Rong Xu** *

Center for Artificial Intelligence in Drug Discovery, School of Medicine, Case Western Reserve University, Cleveland, Ohio, United States of America

* rxx@case.edu

## Abstract

Cancer origin determination combined with site-specific treatment of metastatic cancer patients is critical to improve patient outcomes. Existing pathology and gene expression-based techniques often have limited performance. In this study, we developed a deep neural network (DNN)-based classifier for cancer origin prediction using DNA methylation data of 7,339 patients of 18 different cancer origins from The Cancer Genome Atlas (TCGA). This DNN model was evaluated using four strategies: (1) when evaluated by 10-fold cross-validation, it achieved an overall specificity of 99.72% (95% CI 99.69%-99.75%) and sensitivity of 92.59% (95% CI 91.87%-93.30%); (2) when tested on hold-out testing data of 1,468 patients, the model had an overall specificity of 99.83% and sensitivity of 95.95%; (3) when tested on 143 metastasized cancer patients (12 cancer origins), the model achieved an overall specificity of 99.47% and sensitivity of 95.95%; and (4) when tested on an independent dataset of 581 samples (10 cancer origins), the model achieved overall specificity of 99.91% and sensitivity of 93.43%. Compared to existing pathology and gene expression-based techniques, the DNA methylation-based DNN classifier showed higher performance and had the unique advantage of easy implementation in clinical settings. In summary, our study shows that DNA methylation-based DNN models has potential in both diagnosis of cancer of unknown primary and identification of cancer cell types of circulating tumor cells.

## Introduction

Identification of cancer origins is routinely performed in clinical practice as site-specific treatments improve patient outcomes [1–4]. While some cancer origins are easy to be determined, others are difficult, especially for metastatic and un-differentiated cancer. Cancer origin determination is typically carried out with immunohistochemistry panels on the tumor specimen and imaging tests, which need considerable resources, time, and expense. In addition, pathologic-based procedures have limited accuracy (66–88%) in determining the origins of metastatic cancer [5–8].

Several gene expression- or microRNA-based molecular classifiers have been developed to identify cancer origin. A k-nearest neighbor classifier based on 92 genes showed an accuracy

and optimization to model evaluation. To execute this notebook, the environment needs to be firstly created according to a YAML file available in Github. In addition, we also created a Docker image available in Docker hub (https://hub.docker.com/r/thunder001/cancer_origin_prediction), where you can download it and run the container directly on your computer.

**Funding:** RX and CLZ are funded by the NIH Director's New Innovator Award under the Eunice Kennedy Shriver National Institute of Child Health & Human Development of the National Institutes of Health (DP2HD084068, Xu), NIH National Institute of Aging (R01 AG057557-01, R01 AG061388-01, R56 AG062272-01, Xu) and American Cancer Society Research Scholar Grant (RSG-16-049-01-MPC, Xu). The funders had no role in study design, data collection and analysis, decision to publish, or preparation of the manuscript.

**Competing interests:** The authors have declared that no competing interests exist.

of 84% in identifying primary site of metastatic cancer via cross-validation [9]. Pathwork, a commercially available platform based on similarity score of 1,550 genes between cancer tissue and reference tissue, achieved an overall sensitivity of 88%, an overall specificity of 99% and an accuracy of 89% in identifying tissue of origin [10, 11]. A decision-tree classifier based on 48 microRNA showed an accuracy of 85–89% in identification of cancer primary sites [12, 13], and an updated version, the 64-microRNA based assay, exhibited an overall sensitivity of 85% [14, 15]. A recent support vector machine-based classifier that integrated gene expression and histopathology showed an accuracy of 88% in known origins of cancer samples [16]. Though these molecular platforms have shown better performance in identifying tissue of origin as compared to pathology-based methods, gene expression- or microRNA-bases classifiers are not easy to be implemented in clinic setting partially due to the instability of RNA [17, 18]. In addition, these classifiers have performance of <90% accuracy, which may further limit their wide adoption in clinical settings [17]. Hence, it is desirable to develop higher performance prediction tools for cancer origin determination, which can also be easily implemented in clinical settings.

DNA methylation is a process by which methyl groups are added to the DNA molecule and 70–80% of human genome is methylated [19]. It has been shown that DNA methylation is established in tissue specific manner during development [20, 21]. Though the genomes of cancer patients exhibit overall demethylation, tissue specific DNA methylation markers might be conserved [21]. Indeed, a random forest-based cancer origin classifier using DNA methylation was reported to achieve a performance with 88.6% precision and 97.7% recall in the validation set [18], which demonstrated the usefulness of methylation data in cancer origin prediction. Recently, deep learning technologies have rapidly applied to the biomedical field, including protein structure prediction, gene expression regulation, behavior prediction, disease diagnosis and drug development [22, 23]. Studies show that deep learning-based models often achieved higher performance than traditional machine learning methods (e.g. random forest and support vector machine, etc.) in many settings, such as gene expression inference [24], transcript factor binding prediction [25], protein-protein interaction prediction [26], detection of rare disease-associated cell subsets [27], variant calling [28], clinic trial outcome prediction [29], among others. In this study, we trained and robustly evaluated a high-performance cancer origin predictive model by leveraging the large amount of DNA methylation data available in The Cancer Genome Atlas (TCGA) and the recent developments in deep neural network learning techniques. We demonstrated that our model performed better than traditional pathology- or gene expression-based models as well as methylation-based random forest prediction model.

## Materials and methods

### Datasets

DNA methylation data (Illumina human methylation 450k BeadChip) and clinical information of 8,118 patients across 24 tissue types were obtained from in GDC data portal [30] using TCGAbiolink (Bioconductor package, version 2.5.12) [31]. We excluded six tissue types with less than 100 cases in TCGA to build robust cancer origin classifier. The final data include DNA methylation data and clinical information from 7,339 patients of 18 cancer origins. TCGA data were used for both cancer origin classifier training and evaluation, which were randomly and stratified split into training set (n = 4,403), development set (n = 1,468) and test set (n = 1,468) (Fig 1).

In order to evaluate the classifier trained on TCGA dataset using independent data, we obtained 11 DNA methylation datasets (Illumina 450k platform) from Gene Expression

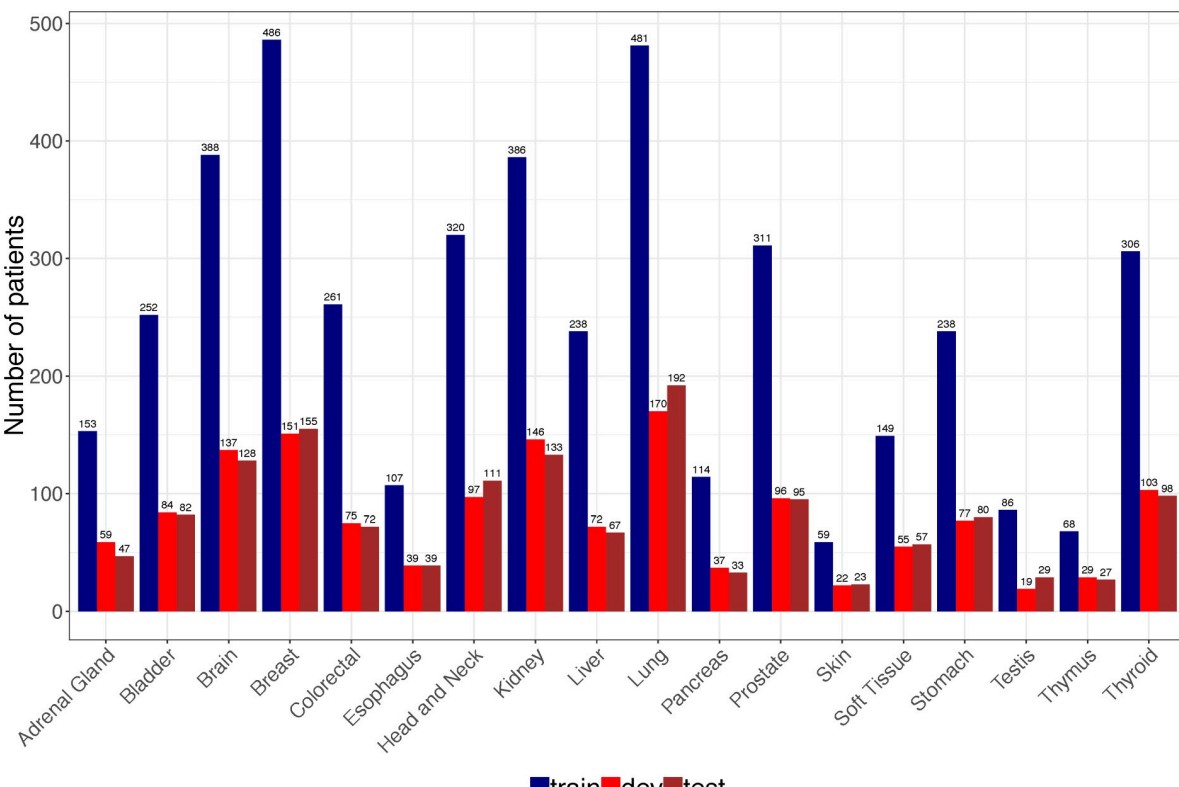

**Fig 1. Distribution of cancer samples in TCGA by tissue of origin.** A total of 7339 patients were randomly and stratified split into train, dev and test sets according to 60:20:20.

Omnibus (GEO) [32] using GEOquery (Bioconductor package, version 2.42.0) [33]. A total of 581 cancer patients covering 10 cancer origins were obtained and the information for each dataset was described in Table 1.

The third DNA methylation data are from 1001 cancer cell lines, which were reported in a large-scale study [34] and deposited in GEO (GSE68379). These cell lines are not treated with

**Table 1. Characteristics of GEO datasets.**

| GEO ID | Disease | Cancer origin | Cancer type | Num. of patients |
|---|---|---|---|---|
| GSE77871 | Adrenocortical carcinomas | Adrenal gland | Primary | 18 |
| GSE78751 | Triple negative breast cancer | Breast | Primary, metastatic | 23 |
| | | | | 12 |
| GSE101764 | Colorectal cancer | Colorectal | Primary | 112 |
| GSE38268 | Head and Neck Squamous Cell Carcinoma | Head and neck | Primary | 6 |
| GSE89852 | hepatocellular carcinomas | Liver | Primary | 37 |
| GSE49149 | Pancreatic cancer | Pancreas | Primary | 167 |
| GSE112047 | Prostate cancer | Prostate | Primary | 31 |
| GSE38240 | Prostate cancer | Prostate | Primary, metastatic | 2 |
| | | | | 6 |
| GSE73549 | Prostate cancer | Prostate | Metastatic | 18 |
| GSE86961 | Papillary thyroid cancer | Thyroid | Primary | 82 |
| GSE52955 | Urology cancer | Kidney, Bladder, prostate | Primary | 17, 25, 25 |

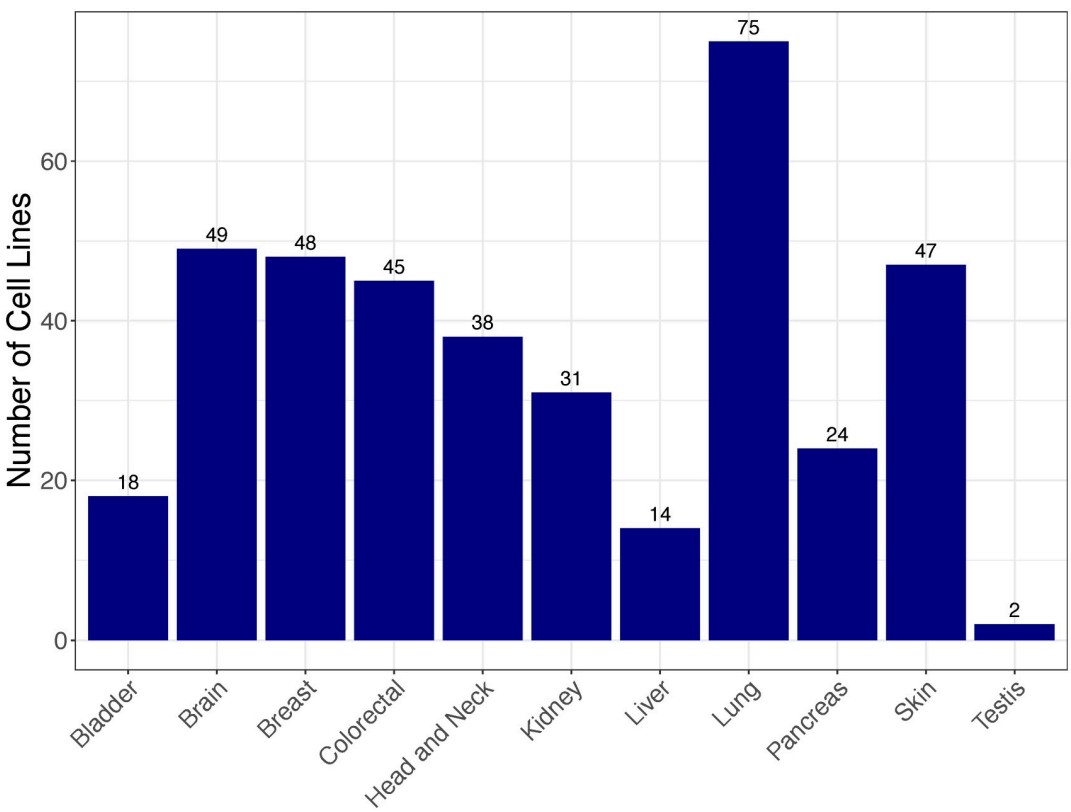

**Fig 2. Distribution of cancer cell lines by tissue source.**

any drug or compound and DNA methylation data were obtained from Illumina 450K Bead-Chip platform. We used this dataset as a case study, i.e., applying our DNN-based tissue classifier to identify the tissue sources of these cancer cell lines. After excluding cancer cell lines whose tissue sources are not covered in our classifier, a total of 391 cell lines from 11 tissue sources were used in this study. Fig 2 shows distribution of these cancer cell lines by tissue sources.

## Feature selection

Only the training data (n = 4,403) from TCGA were used for feature selection. Currently, Illumina 450K and 27K are two commonly used platforms for genome wide analysis of DNA methylation, which measure DNA methylation of around 450K and 27K CpG sites respectively. DNA methylation level of CpG site is expressed as beta value using the ratio of intensities between methylated and unmethylated alleles. Beta value is between 0 and 1 with 0 being unmethylated and 1 fully methylated. All data we used in this study are from 450K platform. In order to reduce the dimensionality while at the same time making the set of features back-compatible with those from 27K platform, we reduced CpG sites to 27K for 450K derived samples by extracting 27K probes from 450K data as the probes used in 450K platform include all probes in 27K platform. To further remove the noise in the data, we used one-way analysis of variance (one-way ANOVA) to filter the CpG sites whose beta values are not significantly different (p > 0.01) among different tissues, resulting in 18,976 CpG sites. Then we used the Tukey's honest significance difference (Tukey's HSD) test to remove the CpG sites that the

maximal difference of their mean beta values among all tissues is less than 0.15. Tukey's HSD is a multiple comparison procedure to find means that are significantly different from each other. It's essentially a t-test that controls family-wise error rate, which is commonly used for post hoc test for ANOVA [35]. The results from Tukey's HSD test are pairwise tissue comparisons with statistical significance and mean difference. Here, we used the pairwise tissue test results to obtain the maximal difference of the mean beta value among all tissues. Tukey's HSD test resulted in 10360 CpG sites that were used for the input layer of neural network.

## Training a deep neural network (DNN) model for cancer origin classification

We used DNA methylation data from training set (n = 4,403) to build a DNN model to predict cancer origins. Tensorflow [36], an open source framework to facilitate deep learning model training, was used for this purpose. Four well-established techniques were used to optimize the training process, including weight initialization by Xaiver method [37], Adam optimization [38], learning rate decay and mini-batch training. Xaiver method can efficiently avoid gradient disappearance/explosion that random initialization may bring. Adam, a combination of Stochastic Gradient Descent with momentum descendent [39] and RMSprop [40], makes training process faster. Exponential learning decay (decay every 1,000 steps with a base of 0.96) was used to improve model performance. Training was performed in 128 mini-batch of 30 epochs to efficiently use the data.

We employed multilayer perceptron (MLP) to construct the neuron network. Three hyperparameters (learning rate, number of hidden layer and hidden layer unit) were optimized according to development set performance (1,468 patients with the same distribution of cancer origins as training set). Three learning rates ($\alpha$ = 0.001, 0.01 and 0.1), three hidden layers (L = 2, 3, 4) and three hidden layer units (N = 32, 64, 128) were tested. We used grid search strategy to optimize these parameters and the best combination according to development set performance is $\alpha$ = 0.001, L = 2 and N = 64.

## Validating and testing DNN-based cancer origin prediction model

We used four strategies to evaluate the performance of the DNN cancer origin classifier: (1) evaluation in the10-fold cross-validation in training dataset to obtain overall specificity, sensitivity, PPV and NPV as well as corresponding confidence intervals of this model; (2) evaluation in the hold-out testing dataset to obtain both the overall model performance and tissue-wise performance; (3) evaluation in the subset of metastatic cancer samples nested in testing dataset to assess the performance of the model in predicting the primary sites of metastatic cancer, which are often more difficult to be identified in clinical practice and more clinically relevant; (4) evaluation in independent datasets from GEO to test the robustness and generalizability of this DNN model. Metrics including specificity, sensitivity, positive predictive value (PPV) and negative predictive value (NPV) were reported. Receiver Operating Characteristic curve (ROC curve) was also calculated for each test data performance.

## Source code, data availability, and reproducibility

Source code used in this study is publicly available in a Github repository (https://github.com/thunder001/Cancer_origin_prediction). We also shared a Jupyter Notebook to replicate all the machine learning experiments from data processing, model building and optimization to model evaluation. To execute this notebook, the environment needs to be firstly created according to a YAML file available in Github. In addition, we also created a Docker image

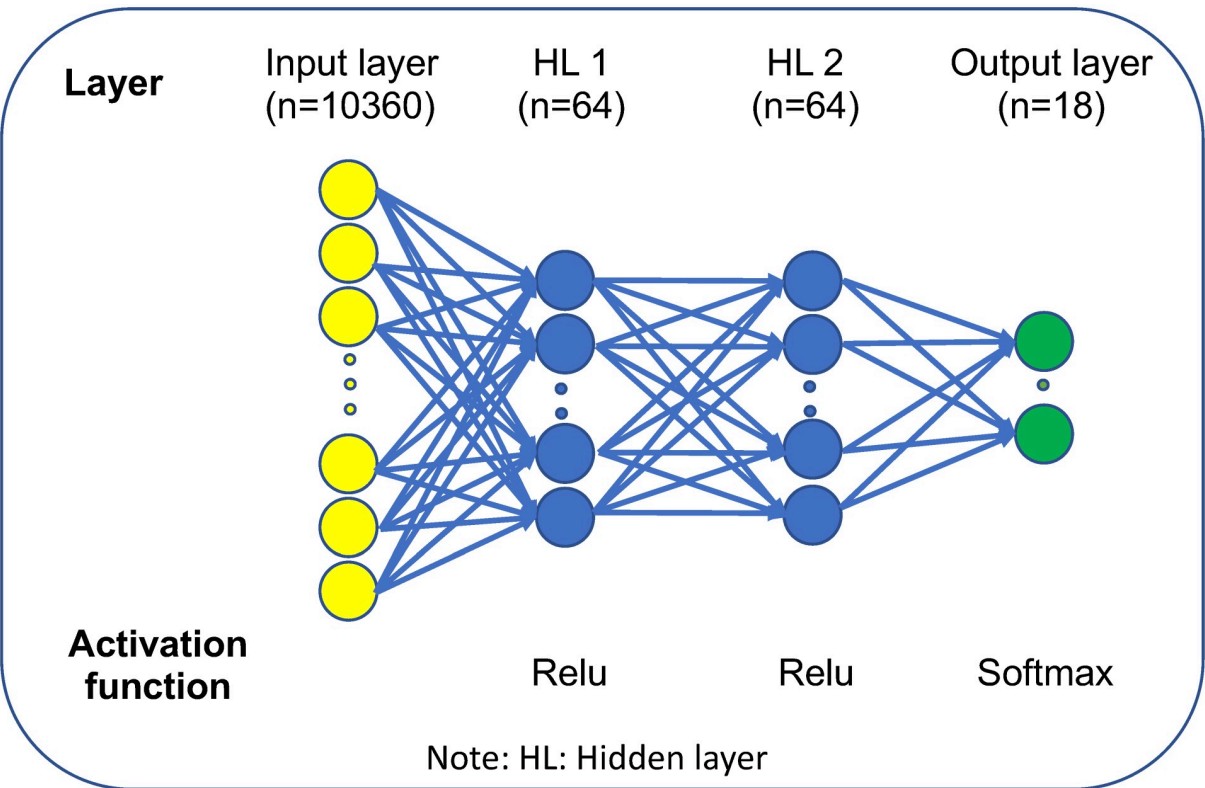

**Fig 3. Schematic representation of DNN architecture of cancer origin classifier.**

available in Docker hub (https://hub.docker.com/r/thunder001/cancer_origin_prediction), where you can download it and run the container directly on your computer.

## Results

### The overall performance of the DNN-based cancer origin classifier in 10-fold cross-validation setting

We used DNA methylation data of 7,339 patients from TCGA across 18 primary tissues to train and test a DNN-based cancer origin classifier. The sample distribution in different cancer origins were shown in Fig 1. The final DNN architecture consists of one input layer (10,360 neurons), two hidden layers (64 neurons each layer) and one output layer (18 neurons) that represents 18 cancer origins (Fig 3).

Evaluated in a 10-fold cross-validation setting, the model achieved an overall precision (positive predictive value, PPV) of 0.9503 (95% CI:0.9373–0.9633) and recall (sensitivity) of 0.9259 (95% CI:0.9187–0.9330) respectively. In addition, this model also achieved a high specificity of 0.9972 (95% CI:0.9969–0.9975) (Table 2).

### DNN-based cancer origin classifier shows high performance in testing dataset

We tested the classifier using test dataset, which includes 1,468 samples with similar distribution with training set (Fig 1). Cancer origin classification and a confusion matrix for all

**Table 2. DNN model performance using 10-fold cross validation of training data.**

|  | Mean | SD | CI (95%) |
|---|---|---|---|
| **Specificity** | 0.9972 | 0.0001 | 0.9969, 0.9975 |
| **Sensitivity (Recall)** | 0.9259 | 0.0032 | 0.9187, 0.9330 |
| **PPV (Precision)** | 0.9503 | 0.0057 | 0.9373, 0.9633 |
| **NPV** | 0.9973 | 0.0001 | 0.9970, 0.9976 |

PPV: positive predictive value; NPV: negative predictive value.

samples were shown in S1 and S2 Tables respectively. Model performance metrics were shown on Table 3. The specificity and negative predictive value (NPV) in individual cancer origin prediction were consistently higher than 0.99. The overall precision (PPV) and recall (sensitivity) reached 0.9608 and 0.9595 respectively. For many cancer tissue origin predictions, including brain, colorectal, prostate, skin, testis, thymus and thyroid, this DNN model achieved a precision of 100% (Table 3) and an average AUC of 0.99 (Fig 4).

There are some variations in precision and recall in different cancer origin predictions. The lowest performance occurred in esophagus origin prediction with a precision of 0.7579 and a recall of 0.7410. A total of 10 of 39 esophagus origins were incorrectly predicted as stomach origins (S1 and S2 Tables). Given that esophagus is a broad area, if a tumor is located at the border of stomach and esophagus, it might be difficult for the classifier to distinguish these two tissues. In addition, tissues from adjacent regions may have similar methylation profiles so that the methylation-based prediction model has difficulty in differentiating cancers with adjacent origins (e.g., esophagus vs stomach).

**Table 3. DNN model performance in test set.**

| CANCER ORIGIN | SPECIFICITY | SENSITIVITY (RECALL) | PPV (PRECISION) | NPV |
|---|---|---|---|---|
| **AG** | 0.9993 | 0.9787 | 0.9787 | 0.9993 |
| **BLADDER** | 0.9986 | 0.9878 | 0.9759 | 0.9993 |
| **BRAIN** | 1.0000 | 1.0000 | 1.0000 | 1.0000 |
| **BREAST** | 0.9977 | 1.0000 | 0.9810 | 1.0000 |
| **COLORECTAL** | 1.0000 | 0.9861 | 1.0000 | 0.9993 |
| **ESOPHAGUS** | 0.9909 | 0.7410 | 0.7579 | 0.9902 |
| **HN** | 0.9971 | 0.9099 | 0.9619 | 0.9927 |
| **KIDNEY** | 0.9993 | 1.0000 | 0.9925 | 1.0000 |
| **LIVER** | 0.9993 | 0.9851 | 0.9851 | 0.9993 |
| **LUNG** | 0.9984 | 0.9740 | 0.9894 | 0.9961 |
| **PANCREAS** | 0.9979 | 1.0000 | 0.9167 | 1.0000 |
| **PROSTATE** | 1.0000 | 1.0000 | 1.0000 | 1.0000 |
| **SKIN** | 1.0000 | 1.0000 | 1.0000 | 1.0000 |
| **SOFT TISSUE** | 0.9993 | 0.9825 | 0.9825 | 0.9993 |
| **STOMACH** | 0.9921 | 0.9375 | 0.8721 | 0.9964 |
| **TESTIS** | 1.0000 | 1.0000 | 1.0000 | 1.0000 |
| **THYMUS** | 1.0000 | 0.8889 | 1.0000 | 0.9979 |
| **THYROID** | 1.0000 | 1.0000 | 1.0000 | 1.0000 |
| **OVERALL** | 0.9983 | 0.9595 | 0.9608 | 0.9983 |

PPV: positive predictive value; NPV: negative predictive value; AG: Adrenal Gland; HN: Head and Neck

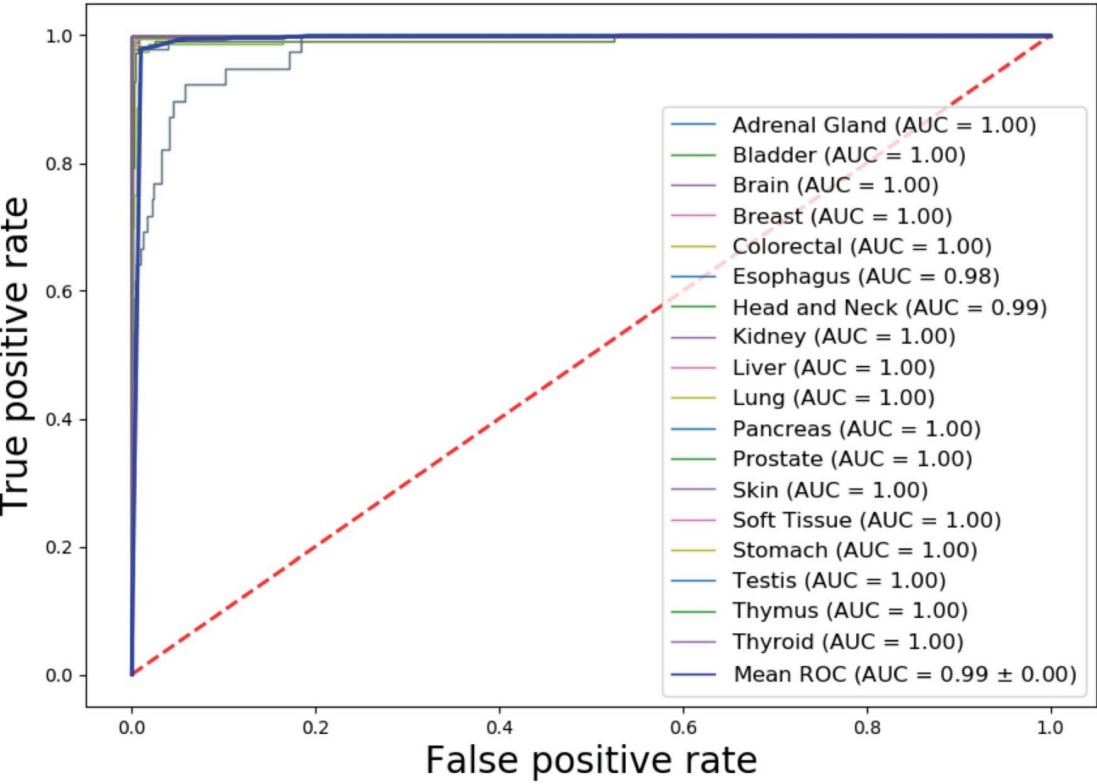

**Fig 4. AUCs for individual cancer origin prediction in TCGA test set.**

## DNN-based cancer tissue classifier shows high performance in determining the origins of metastasized cancers

We evaluated the performance of the classifier in determining the origins of metastatic cancers that nested in our test data. Our data contained 701 samples from distantly metastasized cancers and 558 of them have been used for model development. We then used remaining 143 samples from 12 cancer origins with various sample sizes for evaluation (Fig 5A). Cancer origin predictions and corresponding confusion matrix were shown in S3 and S4 Tables. Model performance metrics and ROC curves were shown in Table 4 and Fig 5B. Consistently, DNN model showed robust high performance in predicting metastatic cancer origins.

We noticed that performance metrics in several cancer origin predictions were poor: a precision of 0.22 for esophagus origin prediction, a precision of 0.67 for liver origin prediction and a recall of 0.67 for lung prediction. The poor performance in these three cancer origin predictions may be due to small sample size. As mentioned above, metastatic cancer samples comprise only a small subset of test dataset in TCGA, the majority of which are primary tumors. Only 2, 2 and 3 metastatic cancer samples from esophagus, liver and lung origin respectively were included in test dataset (Fig 5A). The classifier mis-classified 6 out of 60 head and neck cancers as esophagus origin and 1 of 3 of lung cancers as liver cancers (S4 Table). Due to small sample sizes for esophagus, liver and lung cancers, a few mis-classifications had significant impacts on the precision metrics.

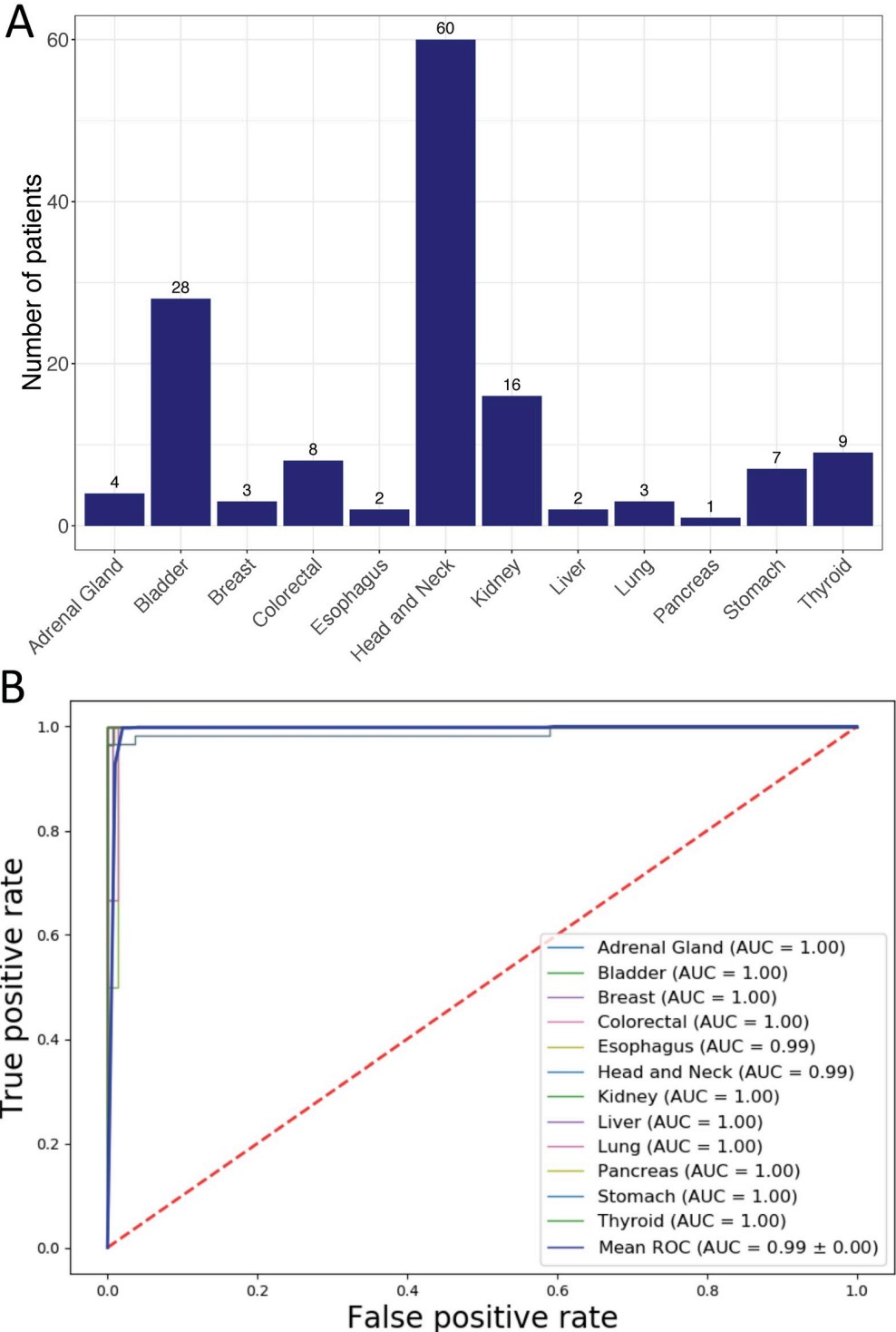

**Fig 5. Performance of the DNN-based cancer origin classifier in metastatic cancer samples from TCGA test set.**
(A) Distribution of metastatic cancer samples by tissue of origin. (B) AUCs for individual cancer origin prediction.

Table 4. DNN model performance in metastatic cancer samples.

| CANCER ORIGIN | SPECIFICITY | SENSITIVITY (RECALL) | PPV (PRECISION) | NPV |
|---|---|---|---|---|
| ADRENAL GLAND | 1.0000 | 1.0000 | 1.0000 | 1.0000 |
| BLADDER | 1.0000 | 0.9643 | 1.0000 | 0.9914 |
| BREAST | 0.9929 | 1.0000 | 0.7500 | 1.0000 |
| COLORECTAL | 1.0000 | 1.0000 | 1.0000 | 1.0000 |
| ESOPHAGUS | 0.9504 | 1.0000 | 0.2222 | 1.0000 |
| HEAD AND NECK | 1.0000 | 0.8833 | 1.0000 | 0.9222 |
| KIDNEY | 1.0000 | 1.0000 | 1.0000 | 1.0000 |
| LIVER | 0.9929 | 1.0000 | 0.6667 | 1.0000 |
| LUNG | 1.0000 | 0.6667 | 1.0000 | 0.9929 |
| PANCREAS | 1.0000 | 1.0000 | 1.0000 | 1.0000 |
| STOMACH | 1.0000 | 1.0000 | 1.0000 | 1.0000 |
| THYROID | 1.0000 | 1.0000 | 1.0000 | 1.0000 |
| OVERALL | 0.9947 | 0.9595 | 0.8866 | 0.9922 |

PPV: positive predictive value; NPV: negative predictive value.

## DNN-based cancer tissue classifier shows high performance in independent testing datasets

The DNN model was trained using DNA methylation data from TCGA. We then tested it in independent datasets of 11 data series consisting of 581 tumor samples covering 10 tissue origins downloaded from Gene Expression Omnibus (GEO). The sample distribution was shown in Fig 6A and cancer origin predictions were listed in S5 Table. Evaluated using these independent datasets, the DNN model achieved high performance with an overall precision and recall of 98.69% and 93.43% respectively (Table 5). High performance was also achieved in individual cancer origin predictions (Table 5) with an average AUC of 0.99 (Fig 6B). Importantly, the model achieved 100% accuracy in predicting the origins of metastatic cancers in these datasets, including 24 prostate cancer that metastasized to bone, lymph node or soft tissue and 12 breast cancer that metastasized to lymph node (see Table 1 for these samples).

## Application of DNN-based cancer tissue classifier in predicting cancer cell type

We next investigate how cancer tissue-trained classifier can be used in cancer cell type prediction. DNA methylation data from 391 cancer cell lines covering 11 tissue sources were obtained from a large-scale study [40]. Applying our classifier into these cancer cell lines, we obtained overall accuracy, precision and recall is 0.8104, 0.8613 and 0.8255 respectively (Table 6). The overall AUC achieves 0.98 (Fig 7). Predicted tissue resource for individual cancer cell line was listed in S7 Table.

Similarly, we noticed variation of model performance for individual cancer cell types. Both precisions and recalls are high in prediction of cancer cell types derived from Brain, Breast, Colorectal, Head and Neck and Skin. However, recall is relatively high in prediction of cancer cell lines from Liver (0.8571) but precision is low (0.4444). Further examining confusion matrix (S8 Table), we found this is caused by mis-prediction of lung cancer cell line as liver cancer cell line. Likewise, precision is high in prediction of pancreatic cell lines (0.9091) but recall is low (0.4167), which is mainly caused by mis-prediction of pancreatic cell lines as stomach and esophagus (S8 Table).

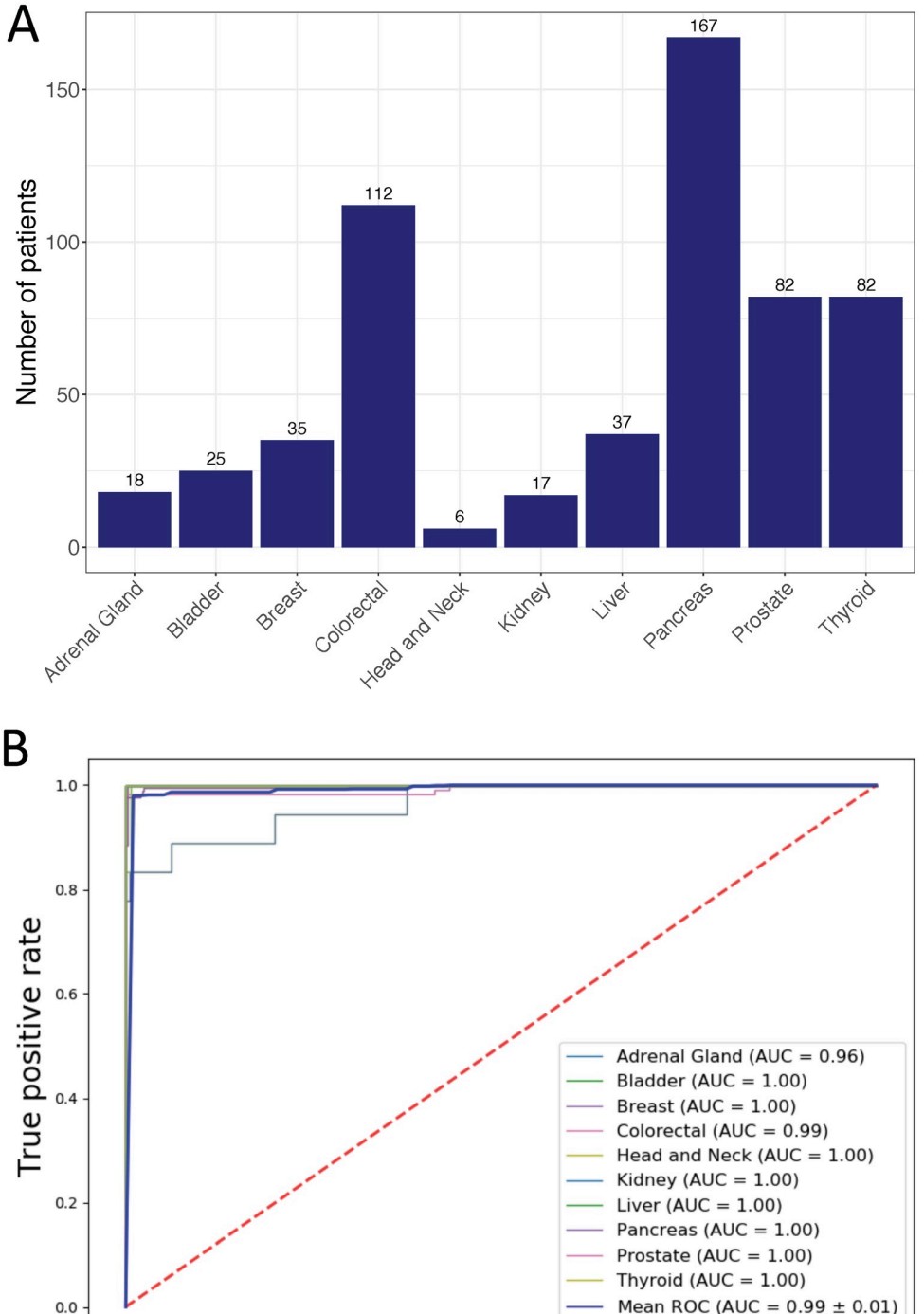

**Fig 6. Performance of the DNN-based cancer origin classifier in GEO dataset.** (A) Distribution of cancer samples obtained from GEO by tissue of origin. (B) AUCs for individual cancer origin prediction.

**Table 5. DNN model performance using independent cancer samples (GEO).**

| CANCER ORIGIN | SPECIFICITY | SENSITIVITY (RECALL) | PPV (PRECISION) | NPV |
|---|---|---|---|---|
| ADRENAL GLAND | 1.0000 | 0.7778 | 1.0000 | 0.9929 |
| BLADDER | 1.0000 | 1.0000 | 1.0000 | 1.0000 |
| BREAST | 0.9963 | 0.9714 | 0.9444 | 0.9982 |
| COLORECTAL | 1.0000 | 0.9643 | 1.0000 | 0.9915 |
| HEAD AND NECK | 1.0000 | 0.8333 | 1.0000 | 0.9983 |
| KIDNEY | 1.0000 | 1.0000 | 1.0000 | 1.0000 |
| LIVER | 0.9945 | 1.0000 | 0.9250 | 1.0000 |
| PANCREAS | 1.0000 | 0.8084 | 1.0000 | 0.9283 |
| PROSTATE | 1.0000 | 1.0000 | 1.0000 | 1.0000 |
| THYROID | 1.0000 | 0.9878 | 1.0000 | 0.9980 |
| OVERALL | 0.9991 | 0.9343 | 0.9869 | 0.9907 |

PPV: positive predictive value; NPV: negative predictive value.

## Discussion

We developed a deep neural network model to predict the cancer origins based on large amount of DNA methylation data from 7,339 patients of 18 different cancer origins. By combining DNA methylation data with deep learning algorithm, our caner origin classifier achieved high performance as demonstrated in four different evaluation settings. Compared with Pathwork, a commercially available cancer origin classifier based on gene expressions [10], our DNN model showed higher precision (95.03% vs 89.4%) and recall (92.3% vs 87.8%) and comparable specificity (99.7% vs 99.4%). Compared with DNA methylation-based random forest model, our DNN model achieved higher PPV (precision) (95.03% in cross validation and 96.08% in test vs 88.6%) and comparable specificity, sensitivity and NPV. In addition, we showed that our DNN model is highly robust and generalizable as evaluated in an independent testing dataset of 581 samples (10 cancer origins), with overall specificity of 99.91% and sensitivity of 93.43%. Therefore, high performance both in primary and metastatic cancer origin prediction and the potential for easy implementation in clinical setting make the methylation-based DNN model a promising tool in determining cancer origins.

**Table 6. DNN model performance in cancer cell type prediction.**

| CANCER CELL TYPE | SPECIFICITY | SENSITIVITY (RECALL) | PPV (PRECISION) | NPV |
|---|---|---|---|---|
| BLADDER | 0.9946 | 0.7778 | 0.8750 | 0.9893 |
| BRAIN | 1.0000 | 0.7959 | 1.0000 | 0.9716 |
| BREAST | 0.9942 | 0.8958 | 0.9556 | 0.9855 |
| COLORECTAL | 0.9942 | 0.9333 | 0.9545 | 0.9914 |
| HEAD AND NECK | 1.0000 | 0.8421 | 1.0000 | 0.9833 |
| KIDNEY | 0.9889 | 0.9355 | 0.8788 | 0.9944 |
| LIVER | 0.9602 | 0.8571 | 0.4444 | 0.9945 |
| LUNG | 0.9842 | 0.6267 | 0.9038 | 0.9174 |
| PANCREAS | 0.9973 | 0.4167 | 0.9091 | 0.9632 |
| SKIN | 0.9826 | 1.0000 | 0.8868 | 1.0000 |
| TESTIS | 0.9974 | 1.0000 | 0.6667 | 1.0000 |
| OVERALL | 0.9903 | 0.8255 | 0.8613 | 0.9810 |

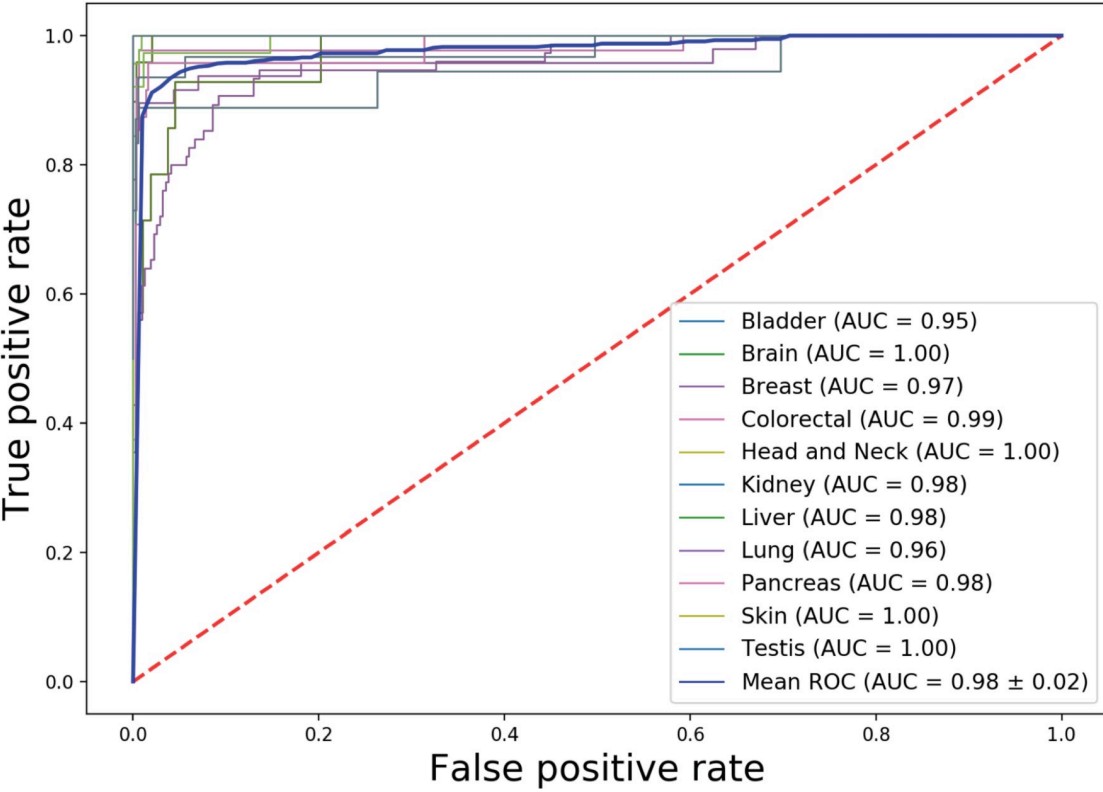

**Fig 7. AUCs for individual cancer cell type prediction.**

DNA methylation is established in tissue specific manner and conserved during cancer development [21], which makes DNA methylation profile a very useful feature in cancer origin prediction. Deep neural networks (DNNs) excels in capturing hierarchical features inherent in many complicated biological mechanisms. Our study indicates that the trained DNN model may be able to capture hierarchical patterns of cancer origins from the DNA methylation data. While Interpretation of deep learning-based models is a rapidly developing field and we expect that our model can be explained in a meaningful way in the future.

Our DNN model has potential in predicting origins of Cancer of Unknown Primary origin (CUP). CUP is a sub-group of heterogenous metastatic cancer with illusive primary site even after standard pathological examination [41]. It is estimated that 3–5% metastatic cancers are CUP and the majority of CUP patients (80%) have poor prognosis with overall survival of 6–10 months [41]. Identifying primary site of CUP poses challenges for treatment decisions in clinical practice. Currently, intensive pathologic examination still leaves 30% of them unidentified [42, 43]. High performance of our DNA methylation-based DNN model may provide an opportunity in this scenario when pathology-based approach fails. However, compared to pathologic examination, DNA methylation-based DNN prediction models has limited interpretability due to the "black-box" nature of deep learning methods and our limited understanding of the mechanistic connections between DNA methylation and cancer origins. We envision that a hybrid approach innovatively combining existing pathological examinations with DNA methylation-based prediction may offer both interpretability and high prediction power.

Due to the limited CUP data in both TCGA and GEO, we currently are unable to test the DNN models in predicting the origins of CUP. Our future direction is to collaborate with hospital to collect DNA methylation data from CUP patients to test our model. One challenge is to obtain the true primary sites for these patients. Due to unknown property of CUP, true primary sites may be established in later cancer development [18]. Another is through the post-mortem examination of patients since 75% of primary sites of CUP were found in autopsy [44].

Another potential usage of our model is to determine the tissue source or cell type of circulating tumor cells (CTCs). CTCs are cells that are shed from primary and metastatic tumors into blood. The enumeration of CTCs is shown an independent prognostic biomarker of overall survival in breast cancer and characteristics of CTCs has shown predictive role of CTCs for patient response to therapy [45, 46]. Role of CTCs in non-invasive diagnosis of cancer also emerges [47, 48]. However, identification its tissue source or tumor type is challenging. Zou J et al has developed eTumorType, which is based on Copy Number Variation (CNV) and shows promising in diagnosis of tumor type of CTCs [49]. We here applied our DNA methylation-based model into cancer cell type identification and our model shows relatively high performance with overall specificity of 0.9903 and overall sensitivity of 0.8255 for 11 cancer types (Table 6). CTCs may have different property from cancer cell lines and we expect that our model can be directly tested in CTCs when sufficient DNA methylation data are available. We are aware that the cell types our model can identified are still limited and performance also varies in different cancer types. Further improvements of our model are warranted.

One limitation of this study is that small sizes of metastatic cancers in our data. Two resources of metastatic cancer were used in this study: TCGA and GEO. TCGA has 701 metastatic cancer samples (12 tissues) with available methylation data from Illumina Human Methylation 450K platform. While the model achieved an overall specificity of 99.47% and sensitivity of 95.95% in cross-validation using TCGA data, we were unable to robustly test it using independent dataset since methylation data of metastatic cancers is limited in GEO. Further independent validation of our DNN-based model in predicting origins of metastatic cancers, especially poorly differentiated or undifferentiated metastatic cancer samples, is needed.

## Conclusion

We developed a DNN-based cancer origin classifier using large-scale of DNA methylation data. This model shows high performance in predicting cancer tissue origins of solid tumors. We also demonstrated the model can be used for cancer cell type identification. In summary, the DNA methylation-based DNN models has potential in diagnosing cancer origin of CUP as well as identifying cancer cell type of CTCs.

## Supporting information

**S1 Table. Cancer origin predictions for 1468 patient samples from TCGA.**
(DOCX)

**S2 Table. Confusion matrix for TCGA test set predictions.**
(CSV)

**S3 Table. Cancer tissue origin predictions for 143 metastatic cancer samples.**
(DOCX)

**S4 Table. Confusion matrix for metastatic cancer samples in TCGA test set.**
(CSV)

**S6 Table. Confusion matrix for GEO sample predictions.**
(CSV)

## Author Contributions

**Conceptualization:** Rong Xu.

**Data curation:** Chunlei Zheng.

**Formal analysis:** Chunlei Zheng.

**Funding acquisition:** Rong Xu.

**Investigation:** Chunlei Zheng, Rong Xu.

**Methodology:** Chunlei Zheng, Rong Xu.

**Project administration:** Rong Xu.

**Resources:** Chunlei Zheng.

**Software:** Chunlei Zheng.

**Supervision:** Rong Xu.

**Validation:** Chunlei Zheng.

**Visualization:** Chunlei Zheng.

**Writing – original draft:** Chunlei Zheng.

**Writing – review & editing:** Rong Xu.

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
