## [Decision Letter · Decision Letter 0]

13 Feb 2020

PONE-D-19-32944

Predicting cancer origins with a DNA methylation-based deep neural network model

PLOS ONE

Dear Dr Xu,

Thank you for submitting your manuscript to PLOS ONE. After careful consideration, we feel that it has merit but does not fully meet PLOS ONE’s publication criteria as it currently stands. Therefore, we invite you to submit a revised version of the manuscript that addresses the points raised during the review process.

We would appreciate receiving your revised manuscript by Mar 29 2020 11:59PM. To enhance the reproducibility of your results, we recommend that if applicable you deposit your laboratory protocols in protocols.io, where a protocol can be assigned its own identifier (DOI) such that it can be cited independently in the future. For instructions see: http://journals.plos.org/plosone/s/submission-guidelines#loc-laboratory-protocols

We look forward to receiving your revised manuscript.

Kind regards,

Serdar Bozdag, Ph.D.

Academic Editor

PLOS ONE

2. To comply with PLOS ONE submission guidelines, in your Methods section, please provide additional information regarding your statistical analyses. For more information on PLOS ONE's expectations for statistical reporting, please see https://journals.plos.org/plosone/s/submission-guidelines.#loc-statistical-reporting.

Reviewers' comments:

Reviewer's Responses to Questions

**Comments to the Author**

1. Is the manuscript technically sound, and do the data support the conclusions?

Reviewer #1: Yes

Reviewer #2: Partly

2. Has the statistical analysis been performed appropriately and rigorously? 

Reviewer #1: Yes

Reviewer #2: Yes

3. Have the authors made all data underlying the findings in their manuscript fully available?

Reviewer #1: Yes

Reviewer #2: Yes

4. Is the manuscript presented in an intelligible fashion and written in standard English?

Reviewer #1: Yes

Reviewer #2: Yes

5. Review Comments to the Author

Reviewer #1: Authors should make a case study which can use the method. For example, CTC methylation information could be used for showing the usefulness of the application. One of the example, CTC was used (PMID: 28389380).

Reviewer #2: General:

The authors proposed a deep neural network-based model to predict origin of cancers using DNA methylation data available in TCGA repository. They divided the dataset into train. development and test sets according to 60:20:20 splits. Other than using 10-fold cross-validation, they also used an independent set of data available in GEO database.

Strength:

• Author clearly mentioned that how cancer site classification study is useful

• Idea of model development and independent validation is good

• Prediction of metastatic samples provides another level of validation

Weakness:

• Major drawback of this model is that it cannot tell which features are responsible for classification or related to the origin of cancers.

• Though the DNN model configuration is mentioned, the author did not specify which version of DNN is used, for example MLP, Autoencoder, etc.

• Design of DNN architecture was not explained clearly. How the number of hidden layers and number of nodes in each hidden layer are decided?

• Feature selection is the key component of this study, which was not explained clearly. Given the information provided, it would be difficult to reproduce the results

• There is no section with “Conclusion.”

Minor Comments:

• Line - 61: “All these molecular platforms have shown better performance in identifying tissue of origin as compared to pathology-based methods.” � This sentence does not make sense with the context of material presented in this section. Needs rephrasing.

• Line-62: “However, gene expression- or microRNA-bases classifiers need to handle RNA that is unstable and less convenient in clinic settings.” �This statement requires some references to support. There is also a spelling mistake, should be … microRNA-based …..

• Line – 114: “Then we used the Tukey honest test to remove….” � Little explanation of “Tukey honest test” and how it works would improve the quality of the paper.

• Line-262: “High performance of our DNA methylation based DNN model may provide an opportunity in this scenario when pathology-based approach fails.” � But it does not provide which DNA methylation sites are related to producing high performance.

Major Comments:

• Need to add a section called “Conclusion”

• Feature Selection: Line-110: “To make the model with good compatibility and also reduce the dimensionality, we firstly reduced CpG sites to 27K for 450K derived samples.” � Concern-1: The authors did not mention what technique was used to reduce the dimension from 450K to 27K. Concern-2: After reducing features 450K to 27K, there will be two sets of 27K features. It is not clear – What do they do with these two sets? Do they combine? Step-by-step procedure to get to the final set of features, 10,360 CpG sites, needs to be stated/explained clearly so that the results can be reproduced.

• Line-127: “In addition, three hyperparameters (learning rate, number of hidden layer and hidden layer unit) were optimized to obtain best performance according to development set performance.” � More details are needed on how each of these hyperparameters are obtained so that the results can be reproduced.

6. PLOS authors have the option to publish the peer review history of their article (what does this mean?). If published, this will include your full peer review and any attached files.

Reviewer #1: Yes: Edwin Wang

Reviewer #2: No

---

## [Decision Letter · Decision Letter 1]

23 Apr 2020

Predicting cancer origins with a DNA methylation-based deep neural network model

PONE-D-19-32944R1

Dear Dr. Xu,

We are pleased to inform you that your manuscript has been judged scientifically suitable for publication and will be formally accepted for publication once it complies with all outstanding technical requirements.

With kind regards,

Serdar Bozdag, Ph.D.

Academic Editor

PLOS ONE

Additional Editor Comments (optional):

Reviewers' comments:

Reviewer's Responses to Questions

**Comments to the Author**

1. If the authors have adequately addressed your comments raised in a previous round of review and you feel that this manuscript is now acceptable for publication, you may indicate that here to bypass the “Comments to the Author” section, enter your conflict of interest statement in the “Confidential to Editor” section, and submit your "Accept" recommendation.

Reviewer #2: All comments have been addressed

Reviewer #3: All comments have been addressed

2. Is the manuscript technically sound, and do the data support the conclusions?

Reviewer #2: Yes

Reviewer #3: (No Response)

3. Has the statistical analysis been performed appropriately and rigorously? 

Reviewer #2: (No Response)

Reviewer #3: (No Response)

4. Have the authors made all data underlying the findings in their manuscript fully available?

Reviewer #2: (No Response)

Reviewer #3: (No Response)

5. Is the manuscript presented in an intelligible fashion and written in standard English?

Reviewer #2: (No Response)

Reviewer #3: (No Response)

6. Review Comments to the Author

Reviewer #2: (No Response)

Reviewer #3: (No Response)

7. PLOS authors have the option to publish the peer review history of their article (what does this mean?). If published, this will include your full peer review and any attached files.

Reviewer #2: No

Reviewer #3: No

---

## [Editor Report · Acceptance letter]

28 Apr 2020

PONE-D-19-32944R1 

Predicting cancer origins with a DNA methylation-based deep neural network model 

Dear Dr. Xu:

I am pleased to inform you that your manuscript has been deemed suitable for publication in PLOS ONE. Congratulations! Your manuscript is now with our production department. 

With kind regards,

on behalf of

Dr. Serdar Bozdag 

Academic Editor

PLOS ONE